# Neurocysticercosis in Ecuador: Spatial clustering, social determinants, and epidemiological trends (2017–2023)

Andrés Fernando Vinueza-Veloz[1]*, Marlon Fabricio Calispa-Aguilar[2], Pamela Vinueza-Veloz[3], Tannia Valeria Carpio-Arias[4], Jefferson Santiago Piedra-Andrade[5], María Fernanda Vinueza-Veloz[6]*, Belkys Galindo-Santana[1]

**1** Department of Epidemiology, Instituto de Medicina Tropical Pedro Kourí, Havana, Cuba, **2** Independent researcher, Quito, Ecuador, **3** Research Group in Veterinary Sciences, Escuela Superior Politécnica del Chimborazo, Riobamba, Ecuador, **4** Research Group on Food and Nutrition (GIANH), Escuela Superior Politécnica del Chimborazo, Riobamba, Ecuador, **5** Escuela de Posgrado, Facultad de Ciencias de Salud, Universidad de las Américas, Quito, Ecuador, **6** Department of Community Medicine and Global Health, Institute of Health and Society, University of Oslo, Oslo, Norway

* andresvinueza1992@gmail.com (AFVV); m.f.v.veloz@medisin.uio.no (MFVV)

## Abstract

### Background

Neurocysticercosis (NCC) is endemic in Ecuador. This study analyzes spatial clustering of municipalities with high and low incidence rates (hot and cold spots) and its association with social determinants of health (SDH).

### Methods

This ecological study used national anonymized hospitalization records (2017–2023) and outpatient registries (2021), as well as municipality-level SDH data. Data were aggregated at the municipality level. Spatial clustering was assessed using Local Indicators of Spatial Association (LISA). SDH were compared between hot and cold spots using parametric and non-parametric tests.

### Results

Between 2017 and 2023, 735 NCC cases were recorded. Annual incidence decreased from 1.01 to 0.44 per 100,000 inhabitants-year between 2017 and 2023. There was no difference in NCC incidence rates between males and females. Spatial analysis revealed clustering of municipalities with high NCC incidence rates in southern Ecuador (Moran's Index = 0.46, $p < 0.001$). In comparison with cold spots, hot spots exhibited higher income inequality (Gini index: 0.48 vs. 0.46; $p < 0.001$), higher density of Creole pigs (6,672.84 vs. 724.73 per 100,000 inhabitants; $p = 0.010$), but lower population density (28.00 vs. 117.20 inhabitants/km²;

**Data availability statement:** NCC cases were extracted from publicly accessible databases: "Hospital Beds and Discharges" for the period 2017 - 2023, which were compiled by the Instituto Nacional de Estadísticas y Censos de Ecuador (INEC), link: https://www.ecuadorenci-fras.gob.ec/camas-y-egresos-hospitalarios/.

**Funding:** The author(s) received no specific funding for this work.

**Competing interests:** The authors have declared that no competing interests exist.

$p < 0.001$), rainfall (1,183.90 vs. 2,126.93 mm³/m²; $p < 0.001$), and temperature (18.47°C vs. 23.98°C; $p < 0.001$). Paradoxically, hot spots had higher sewerage coverage (77.00% vs. 50.70%; $p = 0.010$) and physician density (220.95 vs. 138.69 per 100,000; $p = 0.010$).

## Conclusions

NCC hot spots were identified in southern Ecuador, which included rural, arid and cooler municipalities reliant on traditional pig-rearing and where income distribution was more unequal. Despite available sanitary infrastructure and healthcare access, structural inequalities seem to undermine NCC control efforts in endemic regions. To disrupt NCC persistence, government strategies must prioritize systemic reforms addressing income gaps alongside targeted public health interventions.

## Author summary

Neurocysticercosis (NCC), a parasitic brain infection caused by pork tapeworm larvae, remains a critical health challenge in Ecuador. This study maps where NCC is most concentrated and investigates the social, economic, and environmental factors driving its spread. Using hospital and outpatient records from 2017–2023, we analyzed 735 NCC cases across Ecuador's municipalities. Over this period, annual NCC rates dropped by more than half (from 1.01 to 0.44 cases per 100,000 people), but clusters of high infection rates persisted in southern Ecuador.

These hot spots—characterized as rural, drier, and cooler areas—share two key features: higher income inequality and more backyard pig farming. Paradoxically, these areas also reported higher sewerage coverage and greater density of healthcare providers compared to low-risk areas (referred to as cold spots). This suggests that despite the provision of sanitary infrastructure and healthcare access, structural inequalities continue to undermine efforts to control NCC.

The findings highlight a vicious cycle linking poverty and disease: communities relying on traditional pig-rearing face a heightened risk of NCC, exacerbated by uneven income distribution. To break this cycle, policymakers must address income disparities while simultaneously improving living conditions, healthcare access, and sanitary infrastructure. Integrating these interventions with measures to promote best practices for animal husbandry by smallholder farmers could help Ecuador reduce the burden of this preventable disease. This research underscores that combating NCC is fundamentally a matter of social justice—requiring systemic action to redistribute income and dismantle the structural inequalities that perpetuate poverty and disease.

## Introduction

The flatworm *Taenia solium*, a member of the class Cestoda, shares morphological features with related taenids, including a head (scolex), neck, and a segmented body (strobila) composed of hundreds of proglottids. Its life cycle progresses through three stages: the adult worm, larva (cysticercus), and embryonated egg (oncosphere) [1,2]. The cysticercus contains an invaginated scolex armed with rostellar hooks and four suckers, which facilitate attachment to the intestinal wall of the definitive host (humans) [1]. Adult worms, which reside in the human small intestine, release approximately 20,000 infective eggs daily.

*Taenia solium* transmission begins when eggs excreted in human feces are ingested by intermediate hosts, typically pigs. Humans become accidental intermediate hosts upon ingesting eggs, leading to cysticercosis [3]. After ingestion, eggs hatch and release oncospheres that penetrate the intestinal wall and disseminate via the bloodstream to muscular, ocular, or cerebral tissues, forming cysticerci [3]. Central nervous system involvement termed neurocysticercosis (NCC), occurs in approximately 85% of cysticercosis cases [4].

NCC is the leading cause of acquired epilepsy in endemic regions, and recent studies have associated it with hydrocephalus and migraine headaches [5,6]. Globally, NCC affects 2.56 to 8.30 million people, primarily in endemic areas located in Latin America, Sub-Saharan Africa, and Southeast Asia [7]. In Ecuador, NCC is a public health concern and is responsible for 56,201 disability-adjusted life years (95% CI: 29,961–89,333) [6]. The NCC incidence rate in Ecuador declined from 5 cases per 100,000 inhabitants in 1996 to an annual average of 0.23 cases per 100,000 inhabitants between 2013–2017 [6,8]. Studies carried out after 2017 have examined cysticercosis incidence trends in the country, but have not specifically examined NCC [9].

Social determinants of health (SDH)—defined as the conditions under which people are born, grow, work, live, and age—are strongly linked to the risk of central nervous system infections [10,11]. Lack of access to sanitation infrastructure, social marginalization, and poverty dynamically interact with environmental factors to sustain *Taenia solium* transmission cycles [8,10]. While Ron-Garrido et al. identified municipality-level NCC risk factors, their omission of spatial contrasts between high- and low-risk clusters hinders our understanding on how local drivers intersect with structural inequalities [8].

SDH do not act in isolation, but instead intersect within ecological, economic, and cultural systems to shape disease risk. By dissecting these interlocking systems through spatial contrasts, this study aims at understanding how structural inequities may create layered vulnerabilities, perpetuating cycles where poverty reinforces disease persistence. This study aims to: 1) Describe NCC incidence rate trends from 2017-2023. 2) Identify spatial clustering patterns of NCC incidence. 3) Assess disparities in SDH profiles between high-incidence (hot spots) and low-incidence (cold spots) clusters.

## Methodology

### Ethics statement

The current study used publicly available, anonymized datasets published by Ecuadorian government institutions as well as international non-governmental organizations. No ethical approvals were required as the research exclusively analyzed aggregated, anonymized data accessible under open-access policies. This project was conducted under the framework of the GUAGUA research initiative (Approval Code: IDIPI-257) at Escuela Superior Politécnica de Chimborazo, adhering to institutional ethical guidelines for secondary data analysis.

### Study design and context

This ecological study used national anonymized hospitalization records from 2017 to 2023. To enhance case detection, we also included outpatient registries, though these were only available for 2021. Data on SDH was retrieved from municipality-level records (Table 1). Patient data were aggregated across the 221 municipalities that comprise Ecuador.

**Table 1. Analyzed municipality-level social determinants of health (SDH). For each one, operational definition, unit of measurement, temporal resolution and source are shown. Abbreviations and symbology: km², square kilometer; mm³, cubic millimeter; m², square meter; USD, United States dollar.**

| Social determinant | Operational definition | Unit of measurement | Temporal resolution | Source |
|---|---|---|---|---|
| Drinking water access | Proportion of households with access to piped drinking water within the municipality. | Percentage | 2021 | [13] |
| Sewerage coverage | Proportion of households connected to a sewage system. | Percentage | 2021 | [13] |
| Population density | Number of inhabitants per km². | Inhabitants/km² | 2017 - 2023 | [12] |
| Human Development Index | Composite index that combines education, income, and life expectancy. Higher values indicate higher development. | Index (0–1) | 2016 | [15] |
| Pig density | Number of pigs per 100,000 inhabitants. | Pigs per 100,000 inhabitants | 2021 | [16] |
| Creole pig density | Number Creole pigs per km². | Creole pigs per 100,000 inhabitants | 2021 | [16] |
| Physician density | Number of physicians per 100,000 inhabitants. | Physicians per 100,000 inhabitants | 2020 | [17] |
| Extreme poverty proportion | Proportion of households below the national poverty line (monthly income per person < USD 47.74). | Percentage | 2021 | [18] |
| Income inequality index (Gini coefficient) | Measure of income distribution inequality. Lower values indicate more equality. | Index (0–1) | 2021 | [18] |
| Land occupation index | Proportion of municipal land area occupied by human buildings. | Percentage | 2022 | [19] |
| Municipal temperature | Mean annual temperature. | Degrees Celsius | 2017 - 2023 | [20] |
| Municipal rainfall | Mean annual precipitation. | mm³/m² | 2017 - 2023 | [20] |

These municipalities are grouped into four geographic regions: the Coast region (Pacific coast), the Highlands (Andes mountains), the Amazon (Amazon rainforest), and the Galápagos Islands. In 2017, Ecuador's population was ~ 16.60 million and this increased to ~18.20 million by 2023 [12].

## Data sources

**Cases of neurocysticercosis.** NCC cases were identified from the most recent available national databases: (i) *Camas y egresos hospitalarios* (an anonymized registry of hospitalized patients compiled from public and private hospitals); and (ii) *Consulta externa*, which documents outpatient care encounters [11,12]. While *Camas y egresos hospitalarios* was available from 2017 to 2021, *Consulta externa* was only available for 2021. Both databases include clinical diagnoses coded according to the International Classification of Diseases, 10th Revision (ICD-10). Data were made publicly available by the *Instituto Nacional de Estadísticas y Censos* (INEC) and can be accessed at https://www.ecuadorencifras.gob.ec/camas-y-egresos-hospitalarios/ and https://datosabiertos.gob.ec/dataset/https-almacenamiento-msp-gob-ec-index-php-s-n5kn91dvcgpskpd.

**Social determinants of health.** SDH indicators were extracted from publicly available municipality-level records (Table 1). Data on access to drinking water and sewerage coverage were retrieved from municipality-level reports validated by the *Agencia de Regulación y Control del Agua* [13]. Annual municipality-level population data were derived from the 2022 national census projections [12]. The Human Development Index for each municipality was extracted from a municipality-level report authored by Juan José Illinworth Niemes, following his authorization [14,15].

Economic indicators, including extreme poverty proportion and income inequality index, were extracted from INEC's national household-level *Encuesta de Empleo, Desempleo y Subempleo* [19]. Agricultural indicators, including pig density and Creole pig density were constructed using data from INEC's *Encuesta Continua de Superficie y Producción*

*Agropecuaria* [16]. The physician density indicator was derived from INEC's *Registro de Actividades y Recursos de Salud*, which quantifies healthcare workforce distribution in the Ecuadorian health system [17].

Environmental variables, including rainfall and temperature, were modeled using CHELSA V2.1 (Climatologies at high resolution for the earth's land surface areas), a global meteorological dataset with 1 km² spatial resolution [20]. The land occupation indicator was sourced from Google's Open Buildings Project, a geospatial initiative mapping building footprint in Latin America, Africa, and Southeast Asia [19]. All datasets were harmonized at municipality-level to ensure spatial consistency. All databases are publicly accessible and derived from nationally representative surveys.

**NCC case definition and incidence rate calculation.** NCC cases were defined using the ICD-10 code B69.0 (Cysticercosis of the Central Nervous System). Cases from three municipalities (i.e., El Piedrero, Manga del Cura, and Golondrinas) were excluded due to both unresolved territorial disputes and incomplete SDH indicators. Consequently, the final sample included data from 218 of the 221 Ecuadorian municipalities. Annual incidence rates (cases per 100,000 inhabitants-year) were calculated at municipal, regional, and national levels for the period 2017–2023. Denominators used population estimates derived from projections based on INEC's 2022 national census.

To mitigate Berkson's bias, cases were aggregated by patient´s place of residence (i.e., municipality). For 2021, both outpatient and hospitalized cases were combined in the numerator, with the denominator corresponding to the municipal population for that year. Outpatient data was unavailable for other years; thus, numerators corresponded only to cases identified in *Camas y egresos hospitalarios* (see Methods section). Sex-stratified national incidence rates were calculated using sex-specific numerators (number of male cases, number of female cases) and denominators (male population, female population). Cases from the Galápagos Islands (n = 2) were merged with the Coast region due to geographical proximity and low case numbers.

## Analysis

**Descriptive statistical analysis.** Age and sex distributions were analyzed for combined hospitalized and outpatient cases. Outcomes (fatality rate and hospitalization duration) were reported exclusively for hospitalized cases. Sex-based differences in variables were assessed using Wilcoxon rank-sum tests. National incidence disparities by sex were evaluated via Student's t-tests, and regional variations via analysis of variance (ANOVA). The choice of statistical tests for analyzing numerical variables depended on the distribution of the dependent variable.

**Spatial analysis.** Spatial patterns of NCC were analyzed using annual average municipality-level incidences, visualized through choropleth maps based on geographic polygons (World Geodetic System 84 coordinate system) provided by INEC. To assess spatial relationships, a first-order queen contiguity spatial weights matrix was constructed, where municipalities sharing a border or vertex were defined as neighbors. Geographic centroids of municipalities served as reference points. The Galápagos Islands were excluded due to oceanic isolation, which precluded meaningful adjacency-based analysis.

Global spatial autocorrelation was quantified using Global Moran's Index, with statistical significance determined via Monte Carlo randomization (999 permutations; random seed: 1,234) [21]. A significant positive autocorrelation (Global Moran's Index = 0.46, $p < 0.001$) indicated case clustering. Local spatial clusters were identified using Local Indicators of Spatial Association (LISA) following Anselin's methodology [21].

Hot spots denote geographically concentrated areas of high NCC incidence rates. These were operationally defined as municipalities meeting three criteria: (i) statistically significant positive local Moran's Index values; (ii) incidence rates exceeding the national mean; and (iii) spatial adjacency to municipalities exhibiting similarly elevated rates (high-high clustering). In contrast, cold spots denote geographically concentrated areas of low NCC incidence rates. Cold spots were defined as municipalities with (i) statistically significant positive local Moran's Index values; (ii) incidence rates below the national mean; and (iii) spatial adjacency to municipalities exhibiting similarly low rates (low-low clustering). Statistical significance was evaluated using a Bonferroni correction [21].

SDH variables were compared between hot spots and cold spots using Student's t-tests (for parametric variables) or Wilcoxon rank-sum tests (for non-parametric variables). The choice of test depended on the distribution of the dependent

Diseases

variables. Normality was assessed visually and via Shapiro-Wilk tests (α = 0.05). Analyses were performed using R v4.3 (*tidyverse*) and GeoDa v1.22 for spatial statistics [22,23].

## Results

### NCC Case description and incidence rate

A total of 735 NCC cases were analyzed after excluding cases from municipalities with territorial disputes or unavailability on SDH. Of these, 649 (88.20%) were hospitalized cases, and 86 (11.80%) were outpatient cases. Annual case distribution was as follows: 2017 (122 cases), 2018 (129 cases), 2019 (103 cases), 2020 (60 cases), 2021 (168 cases: 86 outpatient, 82 hospitalized), 2022 (94 cases), and 2023 (59 cases). The lowest incidence corresponded to 2020 (0.34 per 100,000 inhabitants-year) and the highest to 2021 (0.95 per 100,000 inhabitants-year), with a mean incidence of 0.60 per

**National**

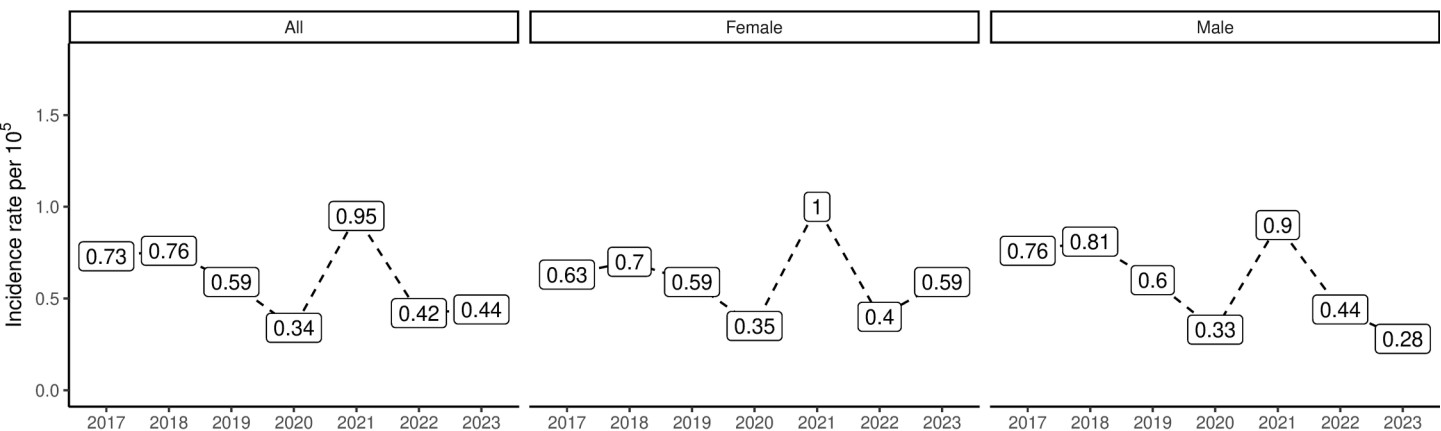

**Regional**

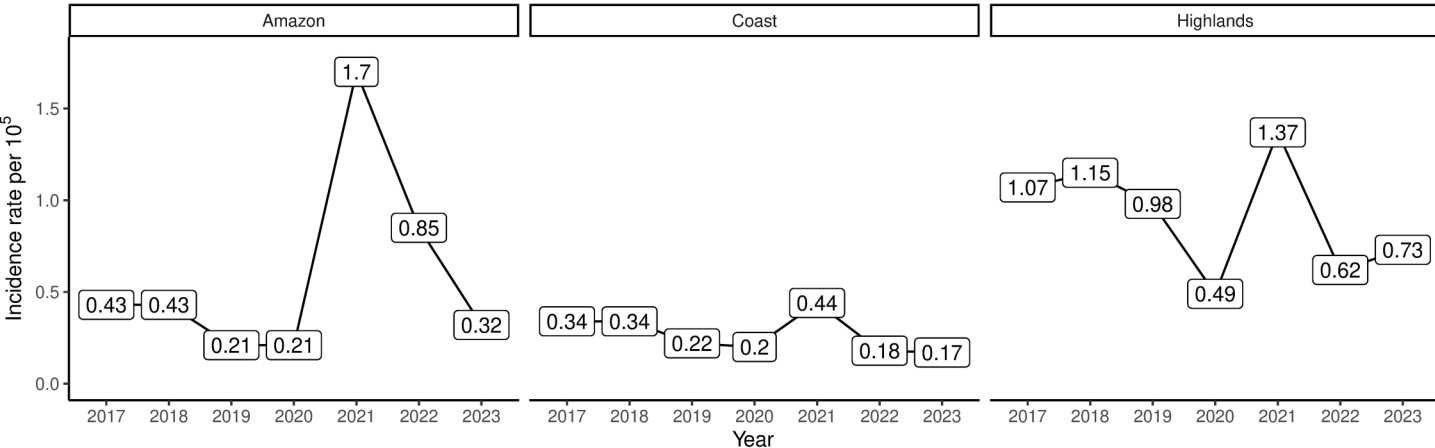

**Fig 1. National and regional neurocysticercosis (NCC) incidence rate by year.** The top panel (National) shows the national incidence rate for females and males combined (All) as well as separated by sex (Female and Male). The bottom panel (Regional) shows the regional incidence rate for each of the geographical regions of Ecuador (Amazon, Coast, and Highlands). The incidence rate for the year 2021 includes cases from hospitalization records and outpatient registries (see Methods).

100,000 inhabitants-year (standard deviation (SD) = 0.21) (Fig 1, National). Sex-specific incidence rates were marginally higher among females (mean = 0.60, SD = 0.21) than males (mean = 0.58, SD = 0.24), though this difference was not statistically significant ($t = 0.16$, $p = 0.874$) (Fig 1, National).

Regional disparities were evident: the Highlands region had the highest mean incidence (0.92 per 100,000 inhabitants-year), followed by the Amazon (0.59 per 100,000 inhabitants-year) and the Coast (0.27 per 100,000 inhabitants-year). The incidence in the Highlands was significantly higher than that of the Coast region ($t = 9.03$, $p < 0.001$) (Fig 1, Regional). Males predominated in outpatient cases and the overall cohort, whereas females constituted the majority of hospitalized cases. No age-related sex differences were observed. Hospitalized cases had a 2.00% fatality rate, with females exhibiting shorter hospitalization duration and slightly higher fatality rates than males, though neither difference reached statistical significance (Table 2).

### Results of spatial analysis

Spatial autocorrelation was detected in the distribution of municipality-level incidences (Global Moran's Index = 0.46, $Z = 11.18$, $p < 0.001$). LISA identified 31 municipalities as significant clusters: 18 cold spots and 13 hot spots. Cold spot municipalities (e.g., Quinindé, Santo Domingo, Portoviejo) were predominantly located in the Coast region, while hot spots (e.g., Cañar, Loja, Palanda) clustered in southern Ecuador (Fig 2). A full list of clustered municipalities is provided in S1 Fig.

Hot spots exhibited significantly higher sewerage coverage (77.00% vs. 50.71%), physician density (220.95 vs. 138.68 per 100,000 inhabitants), income inequality index (0.47 vs. 0.45), and Creole pig density (6,672.84 vs. 724.72 Creole pigs/100,000 inhabitants) compared to cold spots. Additionally, hot spots had lower population density (28.03 vs. 117.25 inhabitants/km²), average temperature (18.47°C vs. 23.98°C), and average rainfall (1,118.90 vs. 2,126.93 mm³/m²) relative to cold spots (Table 3).

### Discussion

NCC remains a significant public health challenge in Ecuador. This study advances the understanding of NCC epidemiology by mapping spatial clustering patterns and identifying SDH that perpetuate the disease burden. Spatial hot spots—located in southern Ecuador—were characterized by lower population density, arid climates, and cooler temperatures compared to spatial cold spots. These high-risk areas exhibited higher income inequality and reliance on traditional pig-rearing, directly linking socioeconomic marginalization and agricultural livelihoods to increased NCC

Table 2. Socio-demographic characterization of the neurocysticercosis (NCC) cases. Combined data included hospitalized and outpatient cases from 2021 (see Methods). Abbreviations and symbology: %, percentage; IQR, interquartile range; n, number; $p$, p-value.

| | Overall | Female | Male | Test |
|---|---|---|---|---|
| **Combined** | | | | |
| Number (%) | 735 | 373 (50.70) | 362 (49.30) | |
| Age, median (IQR) | 47 (32-61) | 48 (33-61) | 43 (31-62) | Wilcoxon test, $p = 0.160$ |
| **Outpatient cases** | | | | |
| Number (%) | 86 | 54 (62.70) | 32 (37.30) | |
| Age, median (IQR) | 47.5 (38-60) | 51.5 (41-60) | 47 (32-63) | Wilcoxon test, $p = 0.350$ |
| **Hospitalized cases** | | | | |
| Number (%) | 649 | 319 (49.10) | 330 (50.90) | |
| Age, median (IQR) | 47 (32-61) | 48 (32-61) | 43 (31-62) | Wilcoxon test, $p = 0.370$ |
| Fatality rate, n (%) | 13 (2.00) | 8 (2.51) | 5 (1.52) | Wilcoxon test, $p = 0.360$ |
| Hospitalization duration, median (IQR) | 5 (3-9) | 5 (2-9) | 6 (3-9) | Wilcoxon test, $p = 0.380$ |

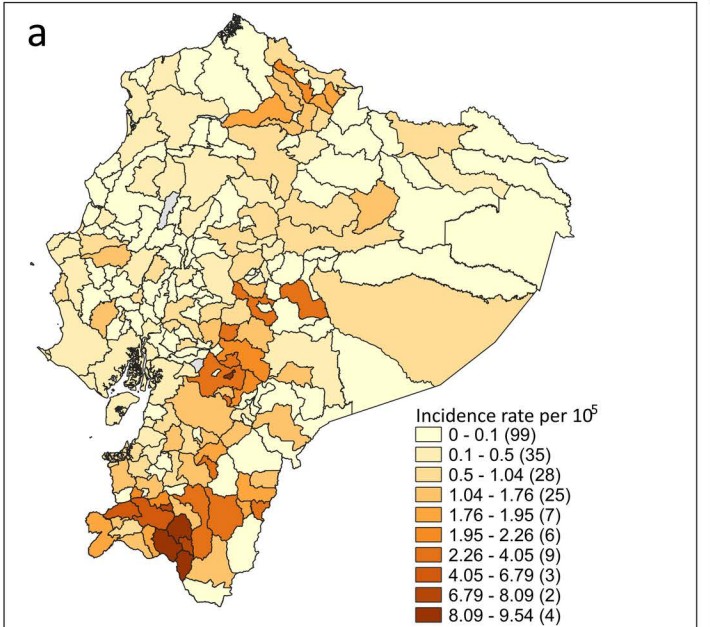

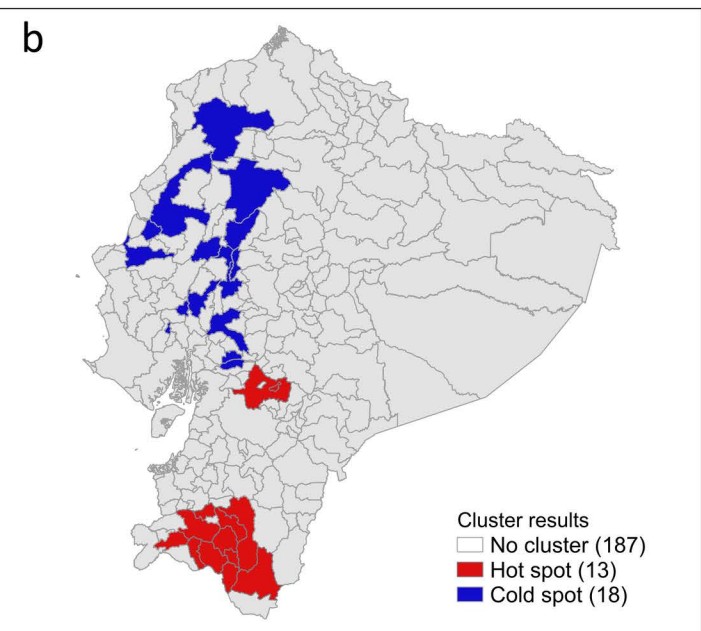

**Fig 2. Geographical distribution of neurocysticercosis (NCC) incidence rates.** The mean municipality-level incidence rates from 2017 to 2023 is shown. **Panel a** shows incidence rates categorized into ten natural breaks. Numbers within the parentheses indicate number of municipalities included in each of the categories. **Panel b** shows the geographical location of hot spots (red) and cold spots (blue). Non-clustered municipalities are shown in gray. Numbers within the parenthesis indicate the number of municipalities included in each of the categories. Base maps sourced from geoBoundaries (ADM-2 shapefile), licensed under CC-BY 4.0 (https://www.geoboundaries.org/countryDownloads.html).

**Table 3. Comparison of social determinants of health (SDH) between hot and cold spots. Except for sewerage coverage and physician density, medians and interquartile ranges are presented. Abbreviations and symbology: *, summary statistics show mean and corresponding standard deviations; df, degrees of freedom; *p*, p-value.**

| Social determinant | Hot spots (n = 13) | Cold spots (n = 18) | Test |
|---|---|---|---|
| Drinking water access | 95.37 (84.06 – 100.00) | 85.81 (58.00 - 94.88) | Wilcoxon test, *p* = 0.110 |
| Sewerage coverage* | 77.00 (28.00) | 50.71 (28.11) | t-test (29 df) = 2.57, *p* = 0.010 |
| Population density | 28.03 (18.55 - 32.44) | 117.25 (57.22 - 168.99) | Wilcoxon test, *p* < 0.001 |
| Land occupation index | 0.81 (0.47 - 1.31) | 1.60 (0.73 - 4.52) | Wilcoxon test, *p* = 0.190 |
| Municipal temperature | 18.47 (14.72 - 18.76) | 23.98 (23.49 - 24.33) | Wilcoxon test, *p* < 0.001 |
| Municipal rainfall | 1,183.90 (1,112.57 – 1,254.16) | 2,126.93 (1,449.26 – 2,779.28) | Wilcoxon test, *p* < 0.001 |
| Human development index | 0.78 (0.77 - 0.78) | 0.77 (0.76 - 0.79) | Wilcoxon test, *p* = 0.360 |
| Pig density | 7,672.10 (1,875.75 – 28,303.53) | 12,107.46 (1,516.60 – 30,473.88) | Wilcoxon test, *p* = 0.560 |
| Creole pig density | 6,672.84 (1,084.86 – 9,410.77) | 724.73 (353.82 – 1,768.10) | Wilcoxon test, *p* = 0.010 |
| Physician density* | 220.95 (85.43) | 138.69 (87.00) | t-test (29 df) = 2.62, *p* = 0.010 |
| Extreme poverty proportion | 13.00 (13.00-13.00) | 8.00 (7.00-9.00) | Wilcoxon test, *p* = 0.050 |
| income inequality index | 0.48 (0.48 - 0.48) | 0.46 (0.43 - 0.47) | Wilcoxon test, *p* < 0.001 |

burden. Paradoxically, hot spots reported higher sewerage coverage and physician density, suggesting that structural inequities counteract gains from improved access to sanitation infrastructure and healthcare. These findings

underscore that NCC persistence reflects systemic failures that demand prioritizing poverty alleviation alongside zoonotic disease control.

### Epidemiological trends: Declining incidence and case ascertainment challenges

Globally, NCC incidence has progressively declined, a trend observed across Latin America [24]. However, methodological inconsistencies hinder cross-study comparisons. For instance, Rodríguez-Rivas et al. grouped NCC cases (ICD-10: B69.0) with cysticercosis in other tissues (ICD-10: B69) in their regional analysis, obscuring NCC case ascertainment [9]. Similarly, while Coral-Almeida et al. reported 170 NCC cases in 2017 using the same ICD-10 code (B69.0) and national database as our study, we only identified 120 cases [6]. This discrepancy likely reflects asynchronous updates across surveillance systems. Standardizing NCC case definitions and harmonizing data protocols are urgently needed to improve surveillance accuracy.

NCC incidence in Ecuador has declined 11-fold since the 1990s (S2 Fig). This decline is likely driven by improvements in the population's living conditions. Key indicators show substantial progress: sewerage access increased from 55% (1990) to 66% (2022), while illiteracy decreased from 12% to 4% over the same period [12,25,26]. Despite these gains, unresolved social inequalities likely contribute to persistent NCC transmission: 5.7 million Ecuadorians lacked sanitation infrastructure access and over half a million remained illiterate in 2022 [12].

Epidemiological studies on NCC in Ecuador, including the current investigation, significantly underestimate disease burden due to reliance on hospital-based case identification [6,8,9,27]. Underreporting is further exacerbated by structural gaps: 12% of Ecuador's population lacks healthcare access, and the country faces a critical shortage of neurologists (0.5 per 100,000 inhabitants vs. the WHO-recommended 1 per 100,000 inhabitants) [28]. Furthermore, fewer than 20% of NCC cases exhibit neurological symptoms that are the primary trigger for clinical detection [5,29]. Consequently, current surveillance systems predominantly capture symptomatic cases, which likely represent the epidemiological "tip of the iceberg", masking the true NCC burden.

The inclusion of outpatient data for the year 2021 offers a closer approximation of the true disease burden and partially mitigates Berkson's bias for that specific year [27]. This methodological adjustment led to a transient spike in NCC incidence (0.95 per 100,000 inhabitants; Fig 1). This peak likely reflects the simultaneous inclusion of both outpatient and hospitalized cases and not a true disease resurgence. Importantly, relying solely on hospitalized data captures fewer than half of cases, even within populations with healthcare access.

Active surveillance strategies can mitigate the underreporting gap for NCC. Strengthening these systems is critical, as precise incidence metrics would enhance the efficacy of preventive interventions. For instance, robust surveillance would enable the implementation of secondary and tertiary prevention strategies to address NCC sequelae, such as epilepsy, stroke, cognitive impairment, and migraine [2,30]. In endemic settings like Ecuador, NCC must be integrated into diagnostic protocols as a secondary etiology for these conditions.

### Sex-based disparities

No sex-based differences were found in overall NCC incidence rates nationwide. However, the percentage of cases skewed toward females (female: 63% vs. male: 37%), whereas inpatient cases showed balanced sex distribution (female: 50% vs. male: 50%). Sex-specific NCC clinical presentations may explain disparities in healthcare utilization. Epileptic seizures, the hallmark manifestation of NCC that affects up to 90% of cases, vary by sex. In this way, males exhibit higher rates of generalized seizures (39% vs. 35% in females), whose overt motor symptoms (e.g., convulsions) heighten urgency for hospitalization [5,31]. In contrast, females more frequently present with complex partial seizures (8% vs. 0% in males), commonly characterized by subtle non-motor symptoms (e.g., language deficits, memory lapses) that often necessitate multiple outpatient evaluations [31].

Sociocultural constructs in the Andean region could further shape these trends. A qualitative study in northwestern Peru—an NCC endemic area—revealed that women, prioritizing caregiving roles, seek early outpatient care to manage symptoms [32]. Conversely, men, whose social identity centers on labor participation, often delay evaluations until symptoms escalate, resulting in hospitalizations [32]. These findings highlight the need for gender-specific surveillance to tailor prevention and care.

### Spatial clustering: Structural inequities, transboundary dynamics, and socioecological persistence

Spatial analysis identified high-incidence NCC clusters (hot spots) in southern Ecuador, characterized by low urban density and subsistence farming reliant on Creole pig-rearing. In contrast, cold spots—clustered in northern regions—were marked by higher industrialization and urban density. The persistence of southern clusters from 1996 to 2023 underscores entrenched socioecological drivers: poverty and marginalization sustain *Taenia solium* transmission [8,10]. Cross-border migration with Peru, where similar subsistence practices prevail, exacerbates endemicity by reintroducing the parasite into vulnerable border zones [33]. This transboundary dynamic, coupled with systemic neglect, necessitates binational surveillance and context-specific interventions to disrupt transmission pathways in persistently marginalized regions.

Paradoxically, hot spots exhibited higher sewerage coverage (77% vs. 51%) and physician density compared to cold spots. These counterintuitive trends may reflect systemic inequalities in infrastructure functionality and healthcare access. In hot spots, despite high reported sewerage coverage, infrastructure often lacks connections to centralized treatment systems. For instance, national data indicate that only 22% of rural households in Ecuador are connected to public sewer systems, with 78% of wastewater discharged untreated into local water bodies [34]. This aligns with studies documenting *Escherichia coli* contamination in 46% of rural drinking water supplies [35]. Disparities in diagnostic capacity further skew reporting: higher physician density in hot spots, combined with clinicians' experience in endemic areas, may inflate case detection rates, while resource limitations in cold spots likely suppress reporting.

Hot spots exhibited higher income inequality, reflecting broader Latin American trends where colonial legacies perpetuate socioeconomic disparities [36,37]. In Ecuador, the colonial past entrenched structural inequities through land concentration and racial hierarchies, displacing Indigenous and marginalized communities into ecologically fragile, less productive lands [36]. These communities often depend on subsistence farming, including rearing Creole pigs, which are perceived as "parasite-resistant," due to their hardiness [38]. This misconception discourages veterinary care-seeking, inadvertently facilitating *Taenia solium* transmission as untreated pigs act as reservoirs [38].

The transmission dynamics of *Taenia solium* are further shaped by environmental and biotic interactions across climatic zones. In arid hot spots, *Taenia solium* eggs face desiccation risks but are protected within the microclimate of human feces, making open defecation—common in sparsely populated rural areas—a critical transmission pathway [39]. In drought-prone southern Ecuador, reliance on untreated wastewater for irrigation exacerbates fecal-oral transmission risks due to potential egg contamination [34,40]. In contrast, cold climate zones such as the Highlands region exhibit distinct risks: backyard-raised Creole pigs, often consumed uninspected during festivals like Carnival, may harbor cysts, perpetuating human exposure [41]. Concurrently, warm and humid conditions in cold spots favor fungal species like *Paecilomyces lilacinus*, which degrade *Taenia solium* eggs, potentially mitigating environmental contamination [39,42]. These regional variations underscore the interplay of climatic factors, human practices (e.g., open defecation, uninspected pork consumption), and biotic agents in modulating disease burden, highlighting the need for context-specific interventions.

To address the persistent NCC burden in southern Ecuador, we recommend prioritizing sanitary control of Creole pork through mandatory veterinary inspections in rural slaughterhouses and informal markets, coupled with vaccination and routine porcine deworming programs [43]. Economic inequities must be mitigated via land redistribution initiatives targeting fertile areas and upgrading irrigation water sanitation systems. Given the 25-year persistence of southern clusters, a binational Ecuador-Peru surveillance system is critical, integrating unified reporting protocols, and cross-border meat contamination controls. Finally, designating NCC as a notifiable disease will enhance epidemiological surveillance, enabling targeted interventions in endemic zones.

## Strengths and limitations

The present study is among the first to apply spatial analysis to identify high-burden NCC clusters and examining associated SDH. This approach provides a contextual framework to better understand NCC persistence. Specifically, we prioritize identifying socio-environmental conditions that perpetuate disease burden rather than quantifying causal pathways. At least two limitations warrant consideration: 1) Reported incidence rates likely underestimate true burden, as case identification relied primarily on hospitalization records. The inclusion of 2021 outpatient data revealed substantial underreporting—case counts increased by ~100% that year, creating an artificial peak in temporal trends. This underestimation is exacerbated by limited healthcare access in remote areas and the subclinical nature of many NCC infections. However, double-counting cannot be ruled out given our use of anonymized public health datasets; patients may transition from outpatient to hospital care without unique identifiers. 2) While incidence data span 2017–2023, SDH indicators were only available for 2021–2023. This temporal mismatch precludes analysis of lagged effects between social determinants and disease trends. However, this limitation aligns with our study's primary objective: identifying spatially contextual risk factors rather than establishing temporal causality.

## Conclusions

NCC transmission persists in geographically clustered hot spots across rural southern Ecuador, characterized by arid climates, cooler temperatures, and economic reliance on traditional pig-rearing. These high-burden municipalities exhibit pronounced income inequality—a structural determinant that undermines NCC control despite nominal gains in sanitation and healthcare access. Our spatial mapping identifies precise intervention targets: government strategies must combine zoonotic disease control with systemic poverty alleviation. Without dismantling the poverty-parasitism cycle, infrastructural or healthcare access improvements alone or educational programs will fail to eliminate NCC.

## Supporting information

**S1 Fig. Spatial location of clusters. Panel a** shows the geographical distribution of the municipalities belonging to hot spots. **Panel b** shows the geographical distribution of the municipalities belonging to cold spots. Base map layers were sourced from geoBoundaries (ADM-2 shapefile), licensed under CC-BY 4.0 (https://www.geoboundaries.org/country-Downloads.html).
(TIF)

**S2 Fig. Neurocysticercosis (NCC) incidence rate in Ecuador from 1996 to 2023.** The figure shows the calculated incidences reported by our team and two other research groups in Ecuador (6,8, this study). The displayed incidences are based on hospitalized cases, with the only exception being the year 2021, which includes both outpatient and hospitalized cases (see Methods section).
(TIF)

## Acknowledgments

To Nelson Gomez, Irma Oliveira, Leda Fernandez and the Neurology Group of the "Hermanos Ameijeiras" Hospital for academic support.

## Author contributions

**Conceptualization:** Andrés Fernando Vinueza-Veloz, Marlon Fabricio Calispa-Aguilar, Pamela Vinueza-Veloz, Tannia Valeria Carpio-Arias, Jefferson Santiago Piedra-Andrade, María Fernanda Vinueza-Veloz, Belkys Galindo-Santana.

**Data curation:** Andrés Fernando Vinueza-Veloz, Marlon Fabricio Calispa-Aguilar, Pamela Vinueza-Veloz, Tannia Valeria Carpio-Arias, Jefferson Santiago Piedra-Andrade, María Fernanda Vinueza-Veloz.

**Formal analysis:** Andrés Fernando Vinueza-Veloz, Marlon Fabricio Calispa-Aguilar, Pamela Vinueza-Veloz, Tannia Valeria Carpio-Arias, María Fernanda Vinueza-Veloz.

**Investigation:** Andrés Fernando Vinueza-Veloz, Marlon Fabricio Calispa-Aguilar, Pamela Vinueza-Veloz, Tannia Valeria Carpio-Arias, Jefferson Santiago Piedra-Andrade, María Fernanda Vinueza-Veloz, Belkys Galindo-Santana.

**Methodology:** Andrés Fernando Vinueza-Veloz, Marlon Fabricio Calispa-Aguilar, Pamela Vinueza-Veloz, Tannia Valeria Carpio-Arias, Jefferson Santiago Piedra-Andrade, María Fernanda Vinueza-Veloz, Belkys Galindo-Santana.

**Project administration:** Andrés Fernando Vinueza-Veloz, Tannia Valeria Carpio-Arias.

**Resources:** Andrés Fernando Vinueza-Veloz, Tannia Valeria Carpio-Arias, Belkys Galindo-Santana.

**Software:** Andrés Fernando Vinueza-Veloz, Tannia Valeria Carpio-Arias.

**Supervision:** Andrés Fernando Vinueza-Veloz, Tannia Valeria Carpio-Arias, María Fernanda Vinueza-Veloz, Belkys Galindo-Santana.

**Validation:** Andrés Fernando Vinueza-Veloz, Marlon Fabricio Calispa-Aguilar, Pamela Vinueza-Veloz, Tannia Valeria Carpio-Arias, Jefferson Santiago Piedra-Andrade, María Fernanda Vinueza-Veloz, Belkys Galindo-Santana.

**Visualization:** Andrés Fernando Vinueza-Veloz, Marlon Fabricio Calispa-Aguilar, Tannia Valeria Carpio-Arias, María Fernanda Vinueza-Veloz.

**Writing – original draft:** Andrés Fernando Vinueza-Veloz, Marlon Fabricio Calispa-Aguilar, Pamela Vinueza-Veloz, Tannia Valeria Carpio-Arias, Jefferson Santiago Piedra-Andrade, María Fernanda Vinueza-Veloz.

**Writing – review & editing:** Andrés Fernando Vinueza-Veloz, Marlon Fabricio Calispa-Aguilar, Pamela Vinueza-Veloz, Tannia Valeria Carpio-Arias, Jefferson Santiago Piedra-Andrade, María Fernanda Vinueza-Veloz, Belkys Galindo-Santana.

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
