## [Decision Letter · Decision Letter 0]

Dear Mr Piedra Andrade,

Thank you very much for submitting your manuscript "Socio-economic and environmental determinants associated with the incidence of neurocysticercosis in Ecuador. Period 2017 – 2022." for consideration at PLOS Neglected Tropical Diseases. As with all papers reviewed by the journal, your manuscript was reviewed by members of the editorial board and by several independent reviewers. In light of the reviews (below this email), we would like to invite the resubmission of a significantly-revised version that takes into account the reviewers' comments.

The reviewers raised concerns about the study's contribution to the existing literature, the lack of acknowledgement of previous similar studies in the introduction, the data sources, figure representations, and the clarity of the English language.

We cannot make any decision about publication until we have seen the revised manuscript and your response to the reviewers' comments. Your revised manuscript is also likely to be sent to reviewers for further evaluation.

Sincerely,

Qu Cheng, Ph.D.

Academic Editor

Cinzia Cantacessi

Section Editor

The reviewers raised concerns about the study's contribution to the existing literature, the lack of acknowledgement of previous similar studies in the introduction, the data sources, figure representations, and the clarity of the English language.

Reviewer's Responses to Questions

**Key Review Criteria Required for Acceptance?**

**Methods**

-Are the objectives of the study clearly articulated with a clear testable hypothesis stated?

-Is the study design appropriate to address the stated objectives?

-Is the population clearly described and appropriate for the hypothesis being tested?

-Is the sample size sufficient to ensure adequate power to address the hypothesis being tested?

-Were correct statistical analysis used to support conclusions?

-Are there concerns about ethical or regulatory requirements being met?

Reviewer #1: This manuscript describes the (spatial)incidence of hospitalised NCC cases in Ecuador between 2017-2022. Additionally, associations are investigated with a number of determinants. However, the latter are rather broad, including multiple factors in one variable (e.g. the poverty by unsatisfied basic needs (PUBN) factor includes adequate housing, availability of electricity, water, sanitation; of some factors which do not necessarily impact on T. solium disease transmission, which render the determinant less informative. A lot of information is available on risk factors, and probably data on water and sanitation, free roaming pigs etc would have been much more relevant. On the level of the determinants, this study brings no new information. It is also not clear why the authors solely focus on hospitalised NCC cases, while from the discussion it becomes clear that outpatient data are available as well (and indicate a much higher number of cases).

However, my main concern with this manuscript is whether it is of added value to the current body of knowledge on NCC in Ecuador. A lot of information is already available. For example, the burden and spatial distribution of NCC in Ecuador has been described for 2013-2017 (Coral-Almeida et al. 2020). Also Rodriguez-Rivas et al. studied the epidemiological situation in Ecuador (1998-2019) including associations with the human development index. The studied period of the latter paper mostly overlaps with the current manuscript (2017-2022), and consults the same database (INEC). The currently published knowledge is completely neglected in the introduction. The latter is very general and a clear rationale for the study is missing.

The authors collected data on the number of pigs, but not on the farming systems? Pigs kept in enclosed, controlled housing systems represent a totally different/no risk for transmission. In the discussion the authors shortly touch upon backyard farming systems. If data is available (and probably it is from previous studies), I would strongly advice to include those in the analyses.

It is not clear whether specific data on access to safe water, presence and use of latrines were collected or only the more broadly covering data PUBN? Why was this not separately collected, that type of data would have been more informative?

From figure 1 it appears that PUBN and HDI were not collected/available from the years from which NCC data were gathered. Can the authors assume that this has no impact on their study? Why did the authors not include NCC data from the years from which they have PUBN and HDI? This is only shortly mentioned in the discussion.

Reviewer #2: I noticed that your study integrates multiple datasets from various sources, each covering different time periods. For example, the hospital discharge records span a five-year period from 2017 to 2022, while other datasets, such as the poverty indicators, are derived from specific years, like the 2010 census. While this method is understood within the field and offers a comprehensive analysis of neurocysticercosis (NCC) incidence and its socio-economic and environmental determinants, it may be beneficial to explicitly clarify this approach for readers who might not be familiar with these nuances. A brief explanation regarding the variation in timeframes could help in addressing any potential complexities in interpreting the results, and assure that the analysis accurately reflects the relationships between these variables over time

Reviewer #3: Well studied and nicely presented. The objective of the study is clearly stated. The study design is appropriate to address the stated objectives. The population clearly described and appropriate for the concerned study. Statistical analysis used to support conclusions. There is NO concern about ethical or regulatory requirements.

**Results**

-Does the analysis presented match the analysis plan?

-Are the results clearly and completely presented?

-Are the figures (Tables, Images) of sufficient quality for clarity?

Reviewer #1: ok

Reviewer #2: The analysis presented in the results section aligns with the study's analysis plan. The results are clearly and comprehensively presented, with appropriate use of statistical measures to support the findings. However, some figures, particularly Figure 2 and Graph 2, could be enhanced for better clarity and resolution. The tables are well-organized and effectively communicate the study's key findings. The spatial distribution of NCC incidence and the identification of high and low incidence clusters are particularly well-executed, adding significant value to the analysis.

Reviewer #3: Results are appropriately presented with analysis.

Figures and tables are appropriate. Figure 3 : should be labelled as A & B for the two pictures. Graphic 1 & 2 should be labelled as Figures and cited appropriately.

**Conclusions**

-Are the conclusions supported by the data presented?

-Are the limitations of analysis clearly described?

-Do the authors discuss how these data can be helpful to advance our understanding of the topic under study?

-Is public health relevance addressed?

Reviewer #1: The correctness of the used databases should be discussed, and the impact this may have on the results.

Here NCC incidence is based on hospitalised cases, while many NCC patients would not have been hospitalised, so here we are looking at the tip of the iceberg? Lines 234-241: here the authors highlight differences in results between their study and data from the MPH based on outpatient consultations: it would be good to not only provide the information, but also reflect on what it means. Or: should the authors include these outpatient data in their study for a better representation of NCC in Ecuador?

More urban than rural hospitalised cases: this is a bit unexpected, and warrants more attention: is there a difference in health seeking behaviour rural versus urban?

Lines 283-292: the authors are advised to revise this paragraph using knowledge specifically for Taenia spp. There is a number of publications describing influence of environmental factors, including climatic, on the survival rates of Taenia spp.

Reviewer #2: The conclusions are generally supported by the data presented. The manuscript successfully discusses the socio-environmental factors influencing NCC incidence and highlights the public health implications of these findings. The limitations of the analysis are mentioned, particularly the use of older poverty data and reliance on hospitalized cases, but these could be expanded upon to provide a more nuanced understanding of the study's constraints. The discussion of how these data can advance our understanding of NCC, especially in the context of low-income countries, is well-handled, though a more explicit focus on public health relevance could strengthen the conclusions.

Reviewer #3: Limitations are identified.

Suggesting to add a clear conclusion statement.

The public health relevance addressed.

**Editorial and Data Presentation Modifications?**

Reviewer #1: The manuscript is written very sloppy, with many mistakes, many abbreviations which makes the manuscript hard to read.

Smaller corrections: (non-exhaustive list, the papers needs a total revision)

the larva u oncosfera

secondary hosts: (accidental)intermediate host

cysticerco: cysticercus

references check, eg line 39: (1, del bruto).

General English: eg In addition, NCC is with 1.2 million disability-adjusted life years loss due to its close association with epilepsy (2) with the more one third of the cases in low- and middle income countries.

The use of many abbreviations complicates the reading

Graph 1, figure 1: this is confusing

Data source: our world in data, what is the reliability of this source? In the reference the correct citation should be used, as presented on the website

Line 228: An in our study??

And many other small typos, eg line 150 sount america…, line 258: during xxx to xxx???

Reviewer #2: The manuscript could benefit from several editorial modifications to enhance clarity. Consistent labeling of figures (e.g., ensuring "Figure 1" is used instead of "Graph 1") is necessary. The resolution of Graph 2 should be improved to match the quality of other figures. Consider revising the title of Figure 1 to explicitly state the "Incidence Rate of Neurocysticercosis in Cantons of Ecuador." Additionally, the authors might consider replacing the photographs in Figure 3 with an infographic that illustrates the life cycle of the Taenia solium parasite, which would provide more educational value and align better with the manuscript's objectives.

Reviewer #3: None identified

**Summary and General Comments**

Reviewer #1: This manuscript describes the (spatial)incidence of hospitalised NCC cases in Ecuador between 2017-2022. Additionally, associations are investigated with a number of determinants. However, the latter are rather broad, including multiple factors in one variable (e.g. the poverty by unsatisfied basic needs (PUBN) factor includes adequate housing, availability of electricity, water, sanitation; of some factors which do not necessarily impact on T. solium disease transmission, which render the determinant less informative. A lot of information is available on risk factors, and probably data on water and sanitation, free roaming pigs etc would have been much more relevant. On the level of the determinants, this study brings no new information. It is also not clear why the authors solely focus on hospitalised NCC cases, while from the discussion it becomes clear that outpatient data are available as well (and indicate a much higher number of cases).

However, my main concern with this manuscript is whether it is of added value to the current body of knowledge on NCC in Ecuador. A lot of information is already available. For example, the burden and spatial distribution of NCC in Ecuador has been described for 2013-2017 (Coral-Almeida et al. 2020). Also Rodriguez-Rivas et al. studied the epidemiological situation in Ecuador (1998-2019) including associations with the human development index. The studied period of the latter paper mostly overlaps with the current manuscript (2017-2022), and consults the same database (INEC). The currently published knowledge is completely neglected in the introduction. The latter is very general and a clear rationale for the study is missing.

Moreover, the manuscript is written very sloppy, with many mistakes, many abbreviations which makes the manuscript hard to read.

Reviewer #2: Overall, this manuscript makes a significant contribution to the understanding of neurocysticercosis in Ecuador. The study's strengths lie in its comprehensive analysis of socio-economic and environmental factors and its robust methodological approach. However, there are areas for improvement, particularly in the presentation of figures and the discussion of limitations. The manuscript’s novelty lies in its detailed examination of NCC clusters in Ecuador, and its significance is underscored by the potential public health applications of the findings. I recommend a major revision to address the identified concerns, particularly regarding the clarity of data presentation and the expansion of the discussion on study limitations.

Reviewer #3: Accepted.

PLOS authors have the option to publish the peer review history of their article (what does this mean? ). If published, this will include your full peer review and any attached files.

**Do you want your identity to be public for this peer review?** For information about this choice, including consent withdrawal, please see our Privacy Policy .

Reviewer #1: No

Reviewer #2: Yes: Esteban Ortiz-Prado

Reviewer #3: Yes: N/A
---

## [Decision Letter · Decision Letter 1]

Response to Reviewers Revised Manuscript with Track Changes Manuscript

Shaden Kamhawi

co-Editor-in-Chief

Paul Brindley

co-Editor-in-Chief

**Reviewers' comments:**

**Key Review Criteria Required for Acceptance?**

**Methods:**

-Are the objectives of the study clearly articulated with a clear testable hypothesis stated?

-Is the study design appropriate to address the stated objectives?

-Is the population clearly described and appropriate for the hypothesis being tested?

-Is the sample size sufficient to ensure adequate power to address the hypothesis being tested?

-Were correct statistical analysis used to support conclusions?

-Are there concerns about ethical or regulatory requirements being met?

Reviewer #2: The objectives of the study are clearly stated, focusing on the spatial distribution of neurocysticercosis (NCC) incidence in Ecuador and its association with socio-environmental determinants. However, the connection between the hypothesis and the methodology, especially regarding the analysis of "hot spots" and "cold spots," could be further clarified to strengthen the link.

The study design is appropriate, utilizing spatial analysis (LISA) and statistical evaluation. However, as it heavily relies on hospital discharge data, the authors should address the potential impact of excluding outpatient data, as it could affect the comprehensiveness of the conclusions.

The population, including hospitalized NCC cases from 2017 to 2023, is clearly defined. The focus on high-incidence clusters is relevant, but a more detailed description of the criteria used for area selection and exclusions would provide additional clarity.

The manuscript does not discuss sample size or power analysis directly, but given the large dataset covering several years and regions, the sample size is likely adequate. It would be beneficial to include a mention of this for transparency and rigor.

The statistical methods, including the use of LISA and regression models, are appropriate. However, the authors should clarify the criteria used to identify "hot spots" and "cold spots" and justify these decisions.

There are no apparent concerns regarding ethical or regulatory requirements, as the manuscript uses publicly available data. However, it would be helpful for the authors to explicitly mention whether ethical approval was obtained or that ethical considerations were reviewed.

**Results:**

-Does the analysis presented match the analysis plan?

-Are the results clearly and completely presented?

-Are the figures (Tables, Images) of sufficient quality for clarity?

Reviewer #2: Does the analysis presented match the analysis plan? Yes, the analysis presented largely matches the outlined plan. The focus on spatial clusters and the investigation of social determinants aligns with the initial objectives. However, as previously mentioned, the exclusion of outpatient data warrants further explanation and may need to be addressed more explicitly in the results.

Are the results clearly and completely presented? The results are generally clear, but there is room for improvement in presenting the findings on the spatial distribution of NCC. The use of clearer labeling and more detailed descriptions of figures, particularly related to the spatial clusters, would improve the clarity of the presentation.

Are the figures (Tables, Images) of sufficient quality for clarity? While the figures and tables are mostly clear, there are concerns with the resolution of some figures (e.g., Figure 2 and Graph 2). Improving the resolution and providing more detailed captions would enhance their clarity and make them more accessible to readers.

**Conclusions:**

-Are the conclusions supported by the data presented?

-Are the limitations of analysis clearly described?

-Do the authors discuss how these data can be helpful to advance our understanding of the topic under study?

-Is public health relevance addressed?

Reviewer #2: Are the conclusions supported by the data presented? The conclusions are largely supported by the data presented. The identification of spatial clusters and the exploration of socio-environmental factors are valid. However, I recommend strengthening the discussion around the implications of the limitations (e.g., reliance on inpatient data) for the overall conclusions.

Are the limitations of analysis clearly described? The limitations are mentioned but could be expanded upon. Specifically, the reliance on inpatient data and the potential impact of using different years for poverty indicators and NCC data should be more thoroughly discussed to provide a more nuanced understanding of the study’s constraints.

Do the authors discuss how these data can be helpful to advance our understanding of the topic under study? Yes, the authors highlight how their findings contribute to understanding the spatial epidemiology of NCC in Ecuador. However, they could strengthen the discussion on how these findings might inform public health interventions, particularly in areas with high pig density or inadequate sanitation.

Is public health relevance addressed? The public health relevance is addressed, particularly in the context of disease control and prevention strategies for NCC in Ecuador. However, a clearer discussion of how these findings can inform policy or intervention strategies would further emphasize the importance of the study.

**Editorial and Data Presentation Modifications?**

Reviewer #2: The manuscript would benefit from several editorial improvements to enhance clarity:

The resolution of some figures, particularly Figure 2 and Graph 2, should be improved for better clarity.

Consistent and accurate labeling of figures should be ensured throughout the manuscript. For example, Figure 1 should clearly indicate that it represents the "Incidence Rate of Neurocysticercosis in Cantons of Ecuador."

Minor typographical errors and awkward phrasing should be corrected for better readability. In particular, the excessive use of abbreviations could be reduced to improve accessibility.

**Summary and General Comments**

Reviewer #2: Overall, this manuscript makes a significant contribution to understanding the epidemiology of neurocysticercosis in Ecuador. The use of spatial analysis and the exploration of socio-environmental determinants are valuable aspects of the study. The study is well-executed but would benefit from further clarification and some methodological adjustments, particularly regarding the use of outpatient data and the population used to calculate incidence rates.

Strengths:

The spatial analysis of NCC incidence and the identification of socio-environmental factors influencing disease distribution are well-executed.

The study provides important insights into the regional and social dynamics of NCC transmission in Ecuador.

Weaknesses:

The reliance on inpatient data may result in an underestimation of the disease burden.

The presentation of the results, particularly the figures and tables, could be improved for clarity.

Further discussion of the limitations, particularly regarding the use of different datasets over time, is needed.

Novelty: The manuscript offers valuable new insights into the spatial epidemiology of NCC in Ecuador, particularly with the extension of the study period and the focus on high-incidence clusters.

Significance: The findings have public health significance, especially in terms of targeting interventions in regions with high NCC incidence.

PLOS authors have the option to publish the peer review history of their article (what does this mean? ). If published, this will include your full peer review and any attached files.

**Do you want your identity to be public for this peer review?** For information about this choice, including consent withdrawal, please see our Privacy Policy .

Reviewer #2: **Yes: ** Esteban Ortiz Prado

**Figure resubmission:****Reproducibility:** To enhance the reproducibility of your results, we recommend that authors of applicable studies deposit laboratory protocols in protocols.io, where a protocol can be assigned its own identifier (DOI) such that it can be cited independently in the future. Additionally, PLOS ONE offers an option to publish peer-reviewed clinical study protocols. Read more information on sharing protocols at https://plos.org/protocols?utm_medium=editorial-email&utm_source=authorletters&utm_campaign=protocols

---

## [Decision Letter · Decision Letter 2]

Response to Reviewers Revised Manuscript with Track Changes Manuscript

Shaden Kamhawi

co-Editor-in-Chief

Paul Brindley

co-Editor-in-Chief

**Reviewers' comments:**

**Key Review Criteria Required for Acceptance?**

**Methods**

-Are the objectives of the study clearly articulated with a clear testable hypothesis stated?

-Is the study design appropriate to address the stated objectives?

-Is the population clearly described and appropriate for the hypothesis being tested?

-Is the sample size sufficient to ensure adequate power to address the hypothesis being tested?

-Were correct statistical analysis used to support conclusions?

-Are there concerns about ethical or regulatory requirements being met?

Reviewer #2: Methods

Are the objectives of the study clearly articulated with a clear testable hypothesis stated?

Yes. The authors present a clearly articulated objective: to estimate the burden of neurocysticercosis (NCC) in Ecuador using national databases. The hypothesis is implicit but testable—that NCC imposes a measurable health burden, particularly in rural areas, based on hospitalization and outpatient consultation data.

Is the study design appropriate to address the stated objectives?

Partially. The use of national databases is appropriate for an ecological burden estimation study. However, there is a significant issue regarding the Consulta externa dataset, which is only publicly available for 2021. This limitation is not clearly stated in the Methods, which may impact the robustness of the outpatient-related findings. Clarifying the temporal scope of this dataset is essential for transparency and replicability.

Is the population clearly described and appropriate for the hypothesis being tested?

Yes. The study includes all registered NCC cases in Ecuador as reported in national databases. The use of population-level data is justified given the ecological nature of the study.

Is the sample size sufficient to ensure adequate power to address the hypothesis being tested?

Yes. The study analyzes all NCC cases recorded in nationwide datasets over several years (for hospitalizations) and at least one year (for outpatient consultations). This ensures sufficient data volume, though clarification on the temporal mismatch is needed.

Were correct statistical analyses used to support conclusions?

Yes. The authors use descriptive and spatial analyses that are standard and appropriate for this type of epidemiological study. However, conclusions drawn from outpatient consultation data must be carefully limited to the available year unless clarified otherwise.

Are there concerns about ethical or regulatory requirements being met?

No. The study uses anonymized, publicly available secondary data from national databases, and no identifiable patient information is used. Ethical standards appear to have been met

**Results**

-Does the analysis presented match the analysis plan?

-Are the results clearly and completely presented?

-Are the figures (Tables, Images) of sufficient quality for clarity?

Reviewer #2: -Does the analysis presented match the analysis plan?

Yes, overall, the analysis appears to follow the stated methodological approach, which includes extracting NCC case data from national-level hospitalization and outpatient consultation databases, followed by descriptive and spatial analysis. However, there is a critical point that requires clarification. The authors cite the Consulta externa database as a continuous source of outpatient care data, but this dataset is only publicly available for the year 2021. This discrepancy between the stated data coverage and actual availability should be explicitly acknowledged in the Methods section to ensure transparency and replicability. Failing to clarify this undermines confidence in the consistency of the outpatient analysis across time and may affect interpretation of trends.

-Are the results clearly and completely presented?

Yes, the results are generally well-structured and clearly presented, with case counts, demographic breakdowns, and geographical patterns detailed both in the text and figures. Nevertheless, given the potential limitation of the outpatient data to a single year, conclusions drawn from these data should be carefully qualified or reinterpreted to avoid overstating temporal trends.

-Are the figures (Tables, Images) of sufficient quality for clarity?

Yes, the figures and tables are of acceptable quality. Maps and tables are legible and appropriately labeled. If the outpatient data are indeed limited to one year, it may be helpful to annotate figures related to outpatient consultations to reflect this and avoid confusion for readers.

**Conclusions**

-Are the conclusions supported by the data presented?

-Are the limitations of analysis clearly described?

-Do the authors discuss how these data can be helpful to advance our understanding of the topic under study?

-Is public health relevance addressed?

Reviewer #2: The conclusions are generally supported by the data presented; however, the limitation regarding the outpatient data (available only for 2021) should be clearly stated. While the authors discuss the broader implications of NCC distribution and its potential public health impact, the analysis would benefit from a more explicit reflection on the data constraints and how they might affect interpretation. Public health relevance is addressed appropriately.

**Editorial and Data Presentation Modifications?**

Reviewer #2: Please clarify the availability of outpatient data used in the analysis. While the source is cited appropriately (Consulta Externa from INEC), it appears that outpatient data are only available for 2021. Since the study spans multiple years, this limitation should be explicitly acknowledged in the methods and discussion sections to ensure transparency and replicability. Aside from this point, the manuscript is clearly written and the figures and tables are of good quality. Minor editorial revisions are suggested for phrasing and consistency. I would recommend Minor Revision pending clarification of the outpatient dataset.

**Summary and General Comments**

Reviewer #2: I appreciate that the authors have addressed my previous comments with clarity and made the necessary adjustments. The manuscript presents a relevant and timely analysis of neurocysticercosis using national data sources in Ecuador, contributing to the understanding of its epidemiology and public health burden.

However, a key methodological clarification remains pending. While the sources of inpatient and outpatient data are described, it appears that outpatient data ("Consulta Externa") is only available for the year 2021. This limitation should be explicitly stated in the Methods section to ensure transparency and allow future replication of the study. Once this issue is addressed, I consider the manuscript close to acceptance with minor revisions.

PLOS authors have the option to publish the peer review history of their article (what does this mean? ). If published, this will include your full peer review and any attached files.

**Do you want your identity to be public for this peer review?** For information about this choice, including consent withdrawal, please see our Privacy Policy .

Reviewer #2: **Yes: ** Esteban Ortiz-Prado

**Figure resubmission:****Reproducibility:** To enhance the reproducibility of your results, we recommend that authors of applicable studies deposit laboratory protocols in protocols.io, where a protocol can be assigned its own identifier (DOI) such that it can be cited independently in the future. Additionally, PLOS ONE offers an option to publish peer-reviewed clinical study protocols. Read more information on sharing protocols at https://plos.org/protocols?utm_medium=editorial-email&utm_source=authorletters&utm_campaign=protocols

---

## [Editor Report · Decision Letter 3]

Dear Dr. Vinueza Veloz,

We are pleased to inform you that your manuscript 'Neurocysticercosis in Ecuador: Spatial clustering, social determinants, and epidemiological trends (2017–2023)' has been provisionally accepted for publication in PLOS Neglected Tropical Diseases.

Best regards,

Qu Cheng, Ph.D.

Section Editor

Cinzia Cantacessi

%CORR_ED_EDITOR_ROLE%

Shaden Kamhawi

co-Editor-in-Chief

Paul Brindley

co-Editor-in-Chief
